



**A particle based model for soil water dynamics: how**
**to match and step beyond Richards' equation?**
Erwin Zehe[1] and Conrad Jackisch[1]
1) Karlsruhe Institute of Technology (KIT)
Abstract:
Within this study we propose a stochastic approach to simulate soil water dynamics in the
unsaturated zone by using a non-linear, space domain random walk of water particles. Soil
water is represented by particles of constant mass, which travel according to the Itô form of
the Fokker Planck equation. The model concept builds on established soil physics by
estimating the drift velocity and the diffusion term based on the soil water characteristics. A
naive random walk, which assumes all water particles to move at the same drift velocity and
diffusivity, overestimated depletion of soil moisture gradients compared to a Richards' solver.
This is because soil water and hence the corresponding water particles in smaller pore size
fractions, are, due to the non-linear decrease of soil hydraulic conductivity with decreasing
soil moisture, much less mobile. After accounting for this subscale variability of particle
mobility, the particle model and a Richards' solver performed similarly during simulated
wetting and drying circles in three distinctly different soils. The particle model typically
produced slightly smaller top soil water contents during wetting and was faster in depleting
soil moisture gradients during subsequent drainage phases. Within a real world benchmark the
particle model matched observed soil moisture response to a moderated rainfall event even
slightly better than the Richards' solver. The proposed approach is hence a promising, easy to
implement alternative to the Richards equation. This is particularly also because it allows one
to step beyond the assumption of local equilibrium during rainfall driven conditions. This is
demonstrated by treating infiltrating event water particles as different type of particle which
travel initially, mainly gravity driven, in the largest pore fraction at maximum velocity, and
yet experience a slow diffusive mixing with the pre-event water particles within a
characteristic mixing time.
Key words: soil water dynamics, random walk, Lagrange model, pre-event water, mobile and
immobile water


## 1    INTRODUCTION

Only a tiny amount of water is stored in the unsaturated zone: with an estimated volume of about 16,500 km$^3$ (Dingman, 1994), soil moisture represents 0.05% of total fresh water. Nevertheless, this tiny storage amount exerts first order control on the partitioning of net radiation energy in latent and sensible heat flux (Kleidon and Renner, 2013a, b; Gayler et al., 2014; Turner et al., 2014) - maybe the key process in land surface atmosphere exchange. Crucially, soil moisture crucially controls $CO_2$ emissions of forest soils (Koehler et al., 2010), de-nitrification and related trace gas emissions into the atmosphere (Koehler et al., 2012) as well as metabolic transformations of pesticides (e.g. Holden and Fierer, 2005). Notwithstanding soil moisture controls splitting of rainfall into surface runoff and (preferential) infiltration (Zehe et al., 2007; Loos and Elsenbeer, 2011; Graeff et al., 2012; Zimmermann et al., 2013; Bronstert et al., 2012; Klaus et al., 2014). Soil water is furthermore a key factor limiting vegetation dynamics in savannah ecosystems (Saco et al., 2007; Tietjen et al., 2010).

Water storage in the unsaturated zone is controlled by capillary forces which increase nonlinearly with decreasing pore size, because water acts as a wetting fluid in soil (Horton and Jury, 2004). The standard approach to represent capillary and gravity controlled soil water dynamics is the Darcy-Richards equation in combination with suitable soil water characteristics. This continuum model essentially assumes that capillarity controlled diffusive fluxes dominate soil water dynamics under local equilibrium conditions even during rainfall driven conditions. Today we know that the assumptions of local equilibrium conditions e.g. (Hassanizadeh et al., 2002; Neuweiler et al., 2012) and a mainly diffusive flow are often not appropriate, particularly during rainfall events in structured soils. Rapid or preferential flow imply a strong local disequilibrium and imperfect mixing between a fast fraction of soil water, travelling in interconnected coarse pores or non-capillary macropores (Šimůnek et al., 2003; Klaus et al., 2013), and the slower diffusive flow in finer fractions of the pore space. As outlined in a couple of excellent review articles (e.g. Šimůnek et al., 2003; Beven and Germann, 2013), up to now many concepts have been proposed to overcome the inability of the Darcy – Richards concept to cope with not-well mixed or even non capillary, preferential flow. These concepts range from a) early stochastic convection (Jury, 1982), b) dual porosity and permeability approaches assuming overlapping and exchanging continua (Gerke and van Genuchten, 1993; van Schaik et al., 2014), to c) spatially explicit representation of



macropores as vertically and laterally connected flow paths (Vogel, 2006; Klaus and Zehe,
2010; Zehe et al., 2010a; Wienhoefer and Zehe, 2014) and d) non local formulations of the
Richards equation (Neuweiler et al., 2012). Notwithstanding the listed short comings, the
Darcy Richards concept works well when soil water dynamics are dominated by capillarity
particularly during radiation driven conditions (Zehe et al., 2010b; Zehe et al., 2014).
Furthermore, it would be foolish to mistake the limitations of the Richards equation with non-
importance of capillary forces in soil. Without capillarity infiltrating rainfall would drain into
groundwater bodies, leaving an empty soil as the local equilibrium state - there would be no
soil water dynamics at all, probably even no terrestrial vegetation and the water cycle would
operate in a complete different manner without capillary forces. Better alternatives for the
Darcy-Richards approach are thus highly desirable, as long they preserve the grain of "truth"
about capillarity as underlying key control.

Here we propose such an alternative approach to simulate soil moisture dynamics in an
effective, stochastic and yet physical way. Specifically, we hypothesise that soil water flow
can be simulated by means of non-linear random walk, representing soil water by a variable
number of particles. To the best of our knowledge, similar Lagrangian approaches were
proposed by Davies and Beven (2012) and taken much further by Ewen (1996b, a). In
accordance with the latter approach our model concept is essentially built on capillarity by
making use of soil physics and established soil water characteristics. Particle tracking based
on a random walk is usually employed for simulating advective-dispersive transport of solutes
in the water phase, but not for the soil water phase itself (Delay and Bodin, 2001; Klaus and
Zehe, 2011; Dentz et al., 2012). For linear problems, when neither the dispersion coefficient
nor the drift term depend on solute concentration and thus particle density, a time domain
representation of the random walk is favourable as it maximises computational efficiency
(Dentz et al., 2012). Non-linear problems, such as transport of nonlinearly adsorbing solutes
or the envisaged simulation of soil water dynamics, require a space domain, random walk,
because the drift and diffusion term change non-linearly with changing particle density. An
integral treatment is, hence, in appropriate as the superposition principle is invalid for non-
linear problems.

In the following we introduce the model concept and present different benchmarks to test its
capability to simulate soil moisture dynamics during equilibrium and non-equilibrium




conditions. More specifically we a) detail the underlying theory and model implementation, b)
reflect on obvious and non-obvious implications of treating water flow in a porous medium as
a non-linear random walk and c) propose a straight forward way to treat non equilibrium
infiltration in section 2. Section 3 explains the model benchmarking a) against a model based
on the Darcy-Richards concept for various soils, initial and boundary conditions as well as b)
against soil moisture observations obtained in the Weiherbach catchment in Germany. After
presenting the results in section 4, we close with discussion and conclusions in section 5.

## 103 2 THEORY AND MODEL IMPLEMENTATION

### 104 2.1 A random walk approach for diffusive water flow in the soil matrix

Our starting point is the equivalence of the Richards equation in the soil moisture based form
to the advection dispersion equation (Jury and Horton, 2004):
$$\frac{\partial \theta}{\partial t} = -k(\theta) + \frac{\partial}{\partial z}\left( D(\theta)\frac{\partial \theta}{\partial z} \right) \quad \text{(Eq.1)}$$

$$D(\theta) = k(\theta)\frac{\partial \psi}{\partial \theta}$$

The volumetric soil water content $\theta[L^3/L^3]$ corresponds to the concentration C $[M/L^3]$ in the
advection diffusion equation; the first term corresponds to a trivial drift/advection term
$u(\theta) = k(\theta)$ [L/T] characterizing downward advective water fluxes driven by gravity. The
second term corresponds to the dispersive/diffusive solute flux, by representing diffusive
water movements driven by the soil moisture gradient and controlled by the diffusivity $D(\theta)$
$[L^2/T]$ of soil water. D is the product of the hydraulic conductivity $k(\theta)$ and the slope of the
soil water retention curve $\frac{\partial \psi}{\partial \theta}$. This equivalence and the work of Ewen (1996 a, b) motivated
the idea to simulate soil water movement by a random walk of a large number of particles.
The soil moisture profile at a given time and within a given spatial discretisation is
represented by the spatial density of "water particles" at this time. Water particles are constant
in mass and volume. The trajectory of a single particle within a time step $\Delta t$ is described by
the Itô form of the Fokker Planck equation:
$$z(t + \Delta t) = -\left( k(\theta(t)) + \frac{\partial D(\theta(t))}{\partial z} \right) \cdot dt + Z\sqrt{6 \cdot D(\theta(t)) \cdot \Delta t} \quad \text{(Eq. 2)}$$



With Z being a random number, uniformly distributed between [1,-1], (note when using
random numbers from the standard normal distribution the drift term is $\sqrt{2 \cdot D(\theta(t)) \cdot \Delta t}$ ). The
term $\partial D(\theta) / \partial z$ corrects the drift term in the case of a spatial variable diffusion as
recommended by (Kitanidis, 1994; Roth and Hammel, 1996; Michalak and Kitanidis, 2000;
Elfeki et al., 2007; Uffink et al., 2012). The main difference to the usual linear random walk is
that D and k depend on soil moisture and thus the water particle density. Here we
parameterise this dependence by means of the van-Genuchten (1980) and Mualem (1976)
model (Figure 1).

### 2.2   Challenges of the particle based approach

#### 2.2.1       Non-linear dependence of D and k on particle density

The obvious implication of the non- linear dependence of the drift velocity and diffusion term
on the soil water content is that a short time stepping in combination with at least a predictor
corrector scheme is needed to account for the non-linear change of both parameters during an
integration time step.

The non-obvious implication arises from the fact that the soil water retention curve reflects
the cumulative pore size distribution of the soil (Jury and Horton, 2004) and the actual soil
moisture reflects water that is stored among different size fractions of the wetted pore space.
It is well known that water flow velocity decreases with decreasing pore size, which is
reflected in the non-linear decrease in soil hydraulic conductivity with decreasing soil water
content. However, this non-linear decrease implies that the water particles representing the
actual soil water content $\theta(t)$ do not all travel at the same constant drift velocity $k(\theta(t))$ and
diffusivity $D(\theta(t))$. In fact only a small fraction of the particles, representing the water in the
largest wetted pores, travels according to these values; the remaining water particles,
representing water stored in smaller pores, are much less mobile.

Hence, the $D(\theta(t))$ and $k(\theta(t))$ curves between the actual soil water content (Figure 1) and
their minimum values may be regarded as a distribution function of the random walk
parameters among the water particles that represent a given soil water content. Essentially, we
propose that a correct random walk implementation needs to account for the different mobility
of the water particles in different pore sizes by resampling the D and k curves from their
minimum to the actual values with a suitable numbers of bins. Contrarily, we expect a naive





execution of Eq. (2), assuming that all particles in a given grid element as equally mobile
according to $k(\theta(t))$ and $D(\theta(t))$, to overestimate fluxes and depletion soil moisture gradients.
### 2.2.2  The necessity to operate at high particle numbers
Another challenge when treating water flow in a Lagrangian approach is that a much larger
number of particles is necessary compared to random walk applications of solute transport.
Why so? The latter treats cases when a solute invades a domain with a small or zero
background concentration of this solute. The total solute mass in the system can thus be
represented by the order of $10^4 - 10^5$ particles even in large, two-dimensional domains at a
good signal-to-noise ratio (Roth and Hammel, 1996; Zehe et al,. 2001). In the case of soil
water dynamics the "background concentration", i.e. the stored pre-event water mass in the
soil profile, is much larger than the input signal of infiltrating event water. The particle
number must thus be considerably increased to the order of $10^6$ in a one dimensional domain,
to ensure that the rainfall input is represented by a number of particles which is sufficiently
high for a stochastic approach.
### 2.3  Equilibrium and non-equilibrium infiltration
Infiltration into the soil at a given $\theta(t)$ is represented as input of event water particles $N^{in}(t)$
into the upper model element, thereby changing the soil water content by $\Delta\theta$. Local
equilibrium conditions, as assumed in the Darcy-Richards concept, imply that water infiltrates
into the smallest non-wetted part of the pore space (as sketched in Figure 1). Consequently the
random walk of the event and pre-event water particles in the largest wetted pores is
determined by $D(\theta(t)+ \Delta\theta)$ and  $k(\theta(t)+ \Delta\theta)$ (Figure 1).

A straightforward approach to account for non-equilibrium infiltration is to assume that event
water enters into and travels in the coarsest pores of the soil, thereby wetting the path of
minimum flow resistance. This implies that diffusive mixing from these coarse pores into the
smallest non-wetted part of the pore space is much slower than the gravity driven downward
flow. Non-equilibrium infiltration may hence be simulated, by assigning the saturated
hydraulic conductivity $k_s$ as drift term "event water particles" and assuming small diffusive
mixing, for instance the lower 5 or 10% quantile of $D(\theta)$. From the latter we specify the time
scales for the event water to mix with the pre-event water as explained further in section 3.2.





### 2.4 Model implementation and execution

#### 2.4.1 Model parameters, initial and boundary conditions

The proposed water particle model is coded in Matlab and requires in its simplest form the same parameters, initial and boundary conditions as a numerical solver of the Richards equation (soil hydraulic functions for the entire soil profile as well as rainfall and optionally evaporation time series). Although the random walk itself does not require a spatial discretisation, we employ a grid to calculate particle densities and soil water contents during run time. The model can be initialized using either an initial soil moisture or matric potential profile for the selected spatial discretisation and based on the selected initial number of water particles $N^{ini}$. The particle mass m [M] is equal to the integral water mass of the initial state divided by N.

Initial positions of the pre-event water particles in a given grid cell are uniformly distributed. Infiltration or soil evaporation is represented as particle input $N^{in}(t)$ or loss $N^{out}(t)$ into/from the upper model element, by dividing the infiltrated/evaporated water mass in a time step by the particle mass. Infiltrating particles start at z=0. Depending on the selected lower boundary condition, particles may either freely drain from the domain (free drainage boundary), a fixed number of particles is kept (constant head boundary), or particles are not allowed to leave the domain (zero flux boundary).

For the implementing non-equilibrium infiltration we treat event water particles as separate type of particles (Figure 1), similar to a different kind of solute that is not influenced by the pre-event water particles unless both fractions are well mixed. Shortly after infiltration we assume event particles to be mainly controlled by gravity; they travel into the vertical according to $k(\theta_s)$ and experience a small diffusive motion characterized by $D_{mix}$. $D_{mix}$ determines the time scale at which pre-event and event water particles get mixed (compare Eq. 3) Non equilibrium implies that the time scale for diffusive mixing $t_{mix}$ is much larger than the time scale of advective transport through a grid element $\Delta z\ t_{ad}$, which implies the grid Peclet number being much larger than 1:



$$\frac{\Delta z k_s}{D_{mix}} = \frac{t_{mix}}{t_{ad}} >> 1$$

$$t_{mix} = \frac{(\Delta z)^2}{D_{mix}}; t_{ad} = \frac{\Delta z}{k_s} \qquad \text{(Eq. 3).}$$


Based on this time scale mixing can be characterised by, for instance, using an exponential
distribution (as proposed by Davies and Beven, 2012). In our study we selected an even
simpler approach, assuming uniformly distributed mixing between the time when the particle
enter the domain and the mixing time. This approach maximises the entropy of the mixing
process (Klaus et al., 2015) thereby minimizing the number of a-priory assumption; because
mixing of each particle is equally likely..

**2.4.2      Time stepping and subscale variability of particle mobility**
For model execution we choose a predictor corrector scheme: we predict the particle
displacement for $0.5*\Delta t$, based on $k(\theta(t))$, $D(\theta(t))$, update $\theta(t+0.5*\Delta t)$ based on the new
particle density distribution and compute the full time step using $k(\theta(t+0.5*\Delta t))$,
$D(\theta(t+0.5*\Delta t))$. As $k(\theta(t))$ and $D(\theta(t))$ are only available at the discrete nodes of the
simulation grid, these are interpolated to the particle locations using inverse distance weights.

We tested two different approaches to cope with the above explained non-linear dependence
of D and k on $\theta(t)$ and thus on particle density. The first, referred to as "full mobility mode",
distributes D among the particles to resemble the shape of D between $D(\theta_r)$ and $D(\theta(t))$ and of
k between $k(\theta_r)$ to $k(\theta(t))$ (Figure 1). To this end we subdivided the particles in a grid cell
representing the actual soil water content $\theta(t)$ and the D and k curves in 800 bins (Figure 1).
This full mobility approach does, however, imply the need to calculate a large chunk of rather
marginal displacements as k and D decline rather fast. The computational less extensive
alternative is to calculate the displacement according to Eq. 2 exclusively for the fastest 10 or
20 % of water particles and assuming the remaining ones to be immobile. Of key interest in
this context is also the question whether the fast mobile and the slow immobile particles
fractions mix across the pores size fractions or not (Brooks et al., 2010). Mixing can be
implemented by assigning the particles randomly to the different bins of during each time step
$D(\theta)$, while no mixing can be realised by always assigning the same particle to same pore size



fraction/ "mobility class". Within our simulations we tested both options. The second option
turned out to be clearly superior with respect to matching simulations with a Richards' solver.

## 3    MODEL BENCHMARKING

### 3.1    Particle model versus Richards equation

In a set of benchmarks we compared the particle model to a numerical solver of the Richards
equation, which was also implemented using Matlab using the same predictor corrector
scheme. We simulated wetting and drying cycles for three soils with rather different soil water
characteristics (Table 1). The first is a sandy soil developed on limestone located in the Attert
experimental basin in Luxembourg (Martinez-Carreras et al., 2010; Wrede et al., 2015). The
second is a young highly porous and highly permeable soil on schistose periglacial deposits in
the Attert basin, which predominantly consists of fine silt aggregates with relative coarse
inter-aggregate pores. The third is a Calcaric Regosol on loess with a large fraction of
medium size pores, which is located at the central meteorological station in the Weiherbach
catchment in south western Germany.

These soils were exposed to simulated wetting and drying cycles characterized in Table 2, by
combining block rains of different intensity with periods of no flux at the upper boundary.
Thereby we compared two different initial soil moisture profiles: a uniform soil water content
of 0.269 $m^3m^{-3}$ and an s-shaped profile. The intensities of block rain events were selected to
be small enough to avoid infiltration excess. Feasible time steps to avoid noisy soil moisture
profiles vary between 60 s for the sand and the young soil on schist and 200 s for the Calcaric
Regosol, when using an initial number of particles of $N^{ini} = 6 * 10^5$. Both models operated at a
constant grid size of 0.05 m in a model domain with a vertical extent of 1.5 m.

### 3.2    Real world benchmark: moderate rainfall event on a loess soil

In the second benchmark we evaluated the particle model against moisture dynamics observed
at the central meteorological station in the Weiherbach catchment (Zehe et al., 2001; Plate and
Zehe, 2008). At this site past rainfall records and soil moisture records in 0.025, 0.1, 0.2, 0.3
and 0.4m are available at a 10 min resolution. We carefully selected a moderate nocturnal
rainfall event, to avoid the influence of macropore flow and evaporation on wetting and
subsequent drying. The event had a total depth of 4 mm with maximum rainfall intensity of 2
mm/h, started at the 9[th] of May at 1:15 and lasted until 4:15 a.m. The changes in soil moisture





in the upper layers revealed a recovery of 90% of the rainfall water, which implies that a
small fraction of the water might have bypassed the sensors.

Both models were operated at a finer spatial discretisation of 0.025 m and we set the number
of pre-event particles to $1*10^6$. The simulation period ranged from 0:05 until 5:45 a.m. at this
day, to allow for a drainage period but to stop simulation before evaporation in the natural
system kicked in. Hydraulic properties of the top and subsoil of the Calcaric Regosol are
given in Table 3. Both models were initialised by assigning the observed soil moisture values,
which increased from 0.18 $m^3m^{-3}$ in 0.025 m to 0.33 $m^3m^{-3}$ in 0.4 m depths, using inverse
distance interpolation between the grid nodes. As no surface runoff occurred during this
event, rainfall was treated as a flux boundary condition.
**4   RESULTS**
In the following we present final soil moisture profiles simulated with the Darcy - Richards
and the particle model for selected runs and compare the temporal evolution of soil moisture
profiles in form of 2d colour plots. In terms of computing time we noted no remarkable
difference between the particle model and the Richards solver. This is because the code is
implemented by relying almost exclusively on array operations, thereby avoiding time-
consuming loops over all particles.
**4.1   Particle model versus Richards equation**
**4.1.1       Sandy soil on lime stone**
Figure 2 presents the final soil moisture profiles for both models for selected simulation
experiments. Panel a) reveals that a treatment of soil moisture dynamics as naïve random
walk, when all particles travel according to $D(\theta(t))$ and $k(\theta(t))$, implies clearly - as expected -
- too fast mixing of event water particles into larger depths compared to the Richards
equation. However, when we accounted for the different mobility of water particles in
different pore sizes, by resembling the distribution of D and k between $D(\theta_r)$ to $D(\theta(t))$ and
$k(\theta_r)$ to $k(\theta(t))$, the particle model closely matched the soil moisture dynamics simulated with
the Richards equation for all simulated wetting cycles. This can be deduced from panels b)
and c) in Figure 2, which show the simulated soil moisture profiles which evolved from a
uniform initial state after a block rain input of 20 and 40 mm, respectively. Panel d) in Figure
2 additionally corroborates the highly similar performance of both models when starting from



the s-shaped initial state. For the sandy soil also we found in general a very good agreement
between the "full mobility" particle model and a simulation assuming a mobile fraction of
20% (solid green line Figure 2 b).

Figure 3 presents the temporal evolution of simulated soil moisture profiles in the form of 2d
colour plots during a 1h block rain of 40 mm (panels a and c) and during a 1h block rain of 20
mm and subsequent drainage of 3h (panels c and d). The accordance between the Richards
solver and the full mobile particle model during rainfall driven conditions was generally high,
regardless of the initial states and rainfall intensities. Small differences occurred in the case of
the 40 mm rainfall, where the particle model produced a slightly larger soil moisture at the
end of the wetting period (Figure 3 panels a) and c)). During non-driven conditions the
particle model was generally faster in depleting soil moisture gradients compared to the
Richards model, as depicted in panels b) and d) in Figure 3.
### 4.1.2      Young silty soil on schist
Simulations of soil water dynamics for the young silty soil on schist, revealed clearer
differences between the Richards equation and the full mobility particle model. The final soil
moisture profiles simulated with the particle model showed smaller values in the upper 20 cm
but larger values between 20 and 40 cm as the corresponding profiles simulated with the
Richards solver (Figure 4, all panels). Differences between simulations of the particle model
operated in the full mobility mode and at a mobile fraction of 20% (Figure 4 panel a) were as
small as in the sandy soil. A better match of the Richards solver required a reduction in the
mobile particle fraction to 10 % (solid green line in panel Figure 4 a). This different behaviour
is likely explained by the larger fraction of medium and small pores in the silty soil compared
to the sandy soil. When simulating a stronger rainfall forcing of 40 mm for 1h both the
Richards' solver and the full mobility particle model were in slightly better accordance as in
the case of the 20 mm input (Figure 4 panel b), while the systematic deviations remained
similar.

The particle model was also more efficient in this soil in depleting sharp soil moisture
contrasts, which became particularly visible when starting with the s-shaped initial soil
moisture profile (Figure 4, panel c). The stronger dissipative character of the particle model
manifested itself even clearer during subsequent drainage periods following on from
infiltration events. This is shown for the case of 20 mm infiltration of the uniform initial state



and subsequent 2 h drainage in the form of 2d colour plots in Figure 5 and for the
corresponding final state in Figure 4 d).

### 4.1.3     Calcaric Regosol on loess

Simulations of soil water dynamics for the Calcaric Regosol on loess were pretty consistent
with those carried out for the young silty soil on schist, particularly with respect to the
systematic differences between the Richards and the particle model. For a total infiltration of
15 mm in 3 h the top 10 to 20 cm, soil moisture simulated with the Richards model was
slightly larger than soil moisture simulated with the full mobility particle model, between 10
to 40 cm it was the other way around (Figure 6 a and b). Again we achieved a better match of
the Richards model when operating the particle model at a mobile fraction of 10%. In general,
differences in model behaviour were more distinct than in the in case of the silty soil on
schist. This is likely explained by the even finer pore sizes in the Calcaric Regosol, which is
reflected in the corresponding air entry values in Table 1.

The finer pore sizes and wider pore size distribution explain also the even stronger dissipative
characters of the full mobility particle model during drainage periods, as corroborated by the
response starting either with the uniform or the s-shaped initial state to 15 mm infiltration in 3
h and 9 h subsequent drainage (Figure 6 c and d). The particle model is faster in redistributing
the water from between the top and the subsoil, which implies smaller soil moisture values
compared to the Richards solver in the upper 0.30 m, but larger soil moisture values between
0.3 and 0.5 m.

### 4.2   Real world benchmark

The real world benchmark in the Calcaric Regosol revealed that the particle model operated in
the full mobility mode performed differently - but not necessarily worse – when compared to
the Richards solver. This can be deduced from the comparison of simulated soil moisture
profiles to observations at the end of the rainfall event (Figure 7 a) and at the end of the
simulation (Figure 7 b). Both models overestimated wetting in 0.025 m, while the particle
model was slightly closer to the observed values at both time steps. The observed wetting of
0.02 $m^3m^{-3}$ in 10 cm at the end of the simulation was well matched by the particle model but
underestimated by the Richards-solver.

A possible explanation for the overestimation of the soil moisture change in 0.025 m by the
models, which is consistent with a non-closed water balance, is that a part of the rainfall water



bypassed the measurement device due fast non-equilibrium infiltration in connected coarse
pores. To test this idea, we performed additional simulations by treating infiltrating event
water particles as a second particle type infiltrating into the largest pores, which uniformly
mixed with the pre-event water particles within the time $t_{mix}$. Figure 7 c) and d) compare the
event water content and total content (as the sum of pre-event and mixed water) for two
different mixing times $t_{mix}$= 4004 ($D_{mix}$ =1.5 $10^{-7}$ $m^2s^{-1}$) and 17144 ($D_{mix}$ =3.3 $10^{-8}$ $m^2s^{-1}$),
which correspond the lower 50 or 30 % quantiles of $D(\theta)$,respectively. Particularly, the model
with the longer mixing time performed distinctly differently to the particle model, assuming
well mixed infiltration. Event water infiltrates and bypasses the pre-event water to a depth of
between 0.1 and 0.3 m in a clearly advective fashion. Related volumetric pre-event water
contents peak at 0.04 $m^3m^{-3}$ (Figure 7 c and d). Consequently, the rainfall input leaves a much
weaker signal in the well mixed water fraction (Figure 8 c), reflecting those event water
particles which diffusively travelled from the coarse pore fraction into the smallest non-
wetted fraction. In case of the faster mixing most of the event water is already mixed with the
pre-event water at the end of the rainfall event (Figure 7 c) and water is completed mixed at
the end of the simulation (Figure 7 d). Consequently, the differences with the simulation
assuming equilibrium infiltration are much less pronounced.

None of the selected mixing time scales did however yield a systematic better performance of
the particle model, in a sense that the mixed water fraction, which we assumed to be in good
contact with the TDR, better matched the observation at 0.025 m depth. This is corroborated
for the final states in Figure 8 c) and d). We thus performed an additional model run assuming
a diffusive mixing according to the 40 % quantile of $D(\theta)$, which corresponds to $t_{mix}$ = 7800s
($D_{mix}$ = 8.8 $10^{-8}$ $m^2s^{-1}$). In this case the simulated well mixed water content was at both times
and in good accordance with the observations at 0.025 m and 0.1 m. We may, hence, state that
the proposed explanation is feasible and that the particle model allows treatment of non-
equilibrium infiltration in a straightforward manner.

## 5    DISCUSSION AND CONCLUSIONS

### 5.1    Subscale variability of water particles – the key to a reasonable performance of non-linear random walk

This study provides evidence that a non-linear, random walk of water particles is a feasible
alternative to the Richards equation for simulating soil moisture dynamics in the unsaturated





zone. The model preserves capillarity as first order control and estimates the drift velocity and
the diffusivity term based on the unsaturated soil hydraulic conductivity and the slope of the
soil water retention curve. As expected, a naive random walk, when all particles in a grid
element travel according to $k(\theta(t))$, $D(\theta(t))$, overestimated depletion of soil moisture gradients
compared to the Richards solver within three different soils for all tested initial and boundary
conditions. The key for improving the particle model performance was to account for the fact
that soil water in different pore size fractions is not equally mobile. When accounting for this
subscale variability in particle mobility in different pore sizes by resampling the D and k
curves from their minimum to the actual values with a suitable numbers of bins, the particle
model performed in good accordance with but yet differently to a Richards solver in three
distinctly different soils. Both models generally gave better accordance during rainfall driven
conditions, regardless of the intensity of the rainfall forcing and the shape of the initial state;
except that the particle model produced slightly smaller top soil water contents. Within
subsequent drying cycles the particle was typically faster in depleting soil moisture gradients
than the Richards' model. This general pattern became more distinct than for a sandy soil
when running models in a finer porous soil consisting of silty aggregates and in a Calcific
Regosol on loess.

### 5.2  Learning about inherent assumption and stepping beyond limitations of the Richards approach

Alternatively, we tested a less computational demanding approach, assuming only the 10 or
20% of the fasted particles to be mobile, while treating the remaining particles located in
smaller pores sizes as immobile. In the case of the sandy soil a mobile fraction of 20%
revealed almost identical results as the full mobility model and performed even closer to the
Richards solver. In the two closer soils the mobile fraction mode was generally less dispersive
then the full mobility model and again in better accordance with the Richards solver,
particularly when setting the mobile particle fraction to 10%. In this context we compared
also the cases of perfect mixing and no mixing between mobile and immobile water particles
between different time steps (as explained in section 2.4.2). The second option was clearly
superior with respect to matching simulations with a Richards' solver, while the other yielded
strong differences. We may thus state that the particle model is a suitable tool to "unmask" a)
inherent implications of the Darcy-Richards concept on the fraction of soil water that actually
contributes to soil water dynamics and b) the inherent very limited degrees of freedom for
mixing between mobile and immobile water fractions. Our findings suggest, furthermore, that





the idea of two separate water worlds, one supplying runoff the other supplying transpiration,
which is advocated in Brooks et al. (2010), is a somewhat naïve interpretation of soil physics
and the inherently low degrees of freedom water to mix across pores size fractions, than a real
mystery.

In a real world benchmark the particle model matched observed soil moisture reactions in
response to a moderate rainfall event even better than the Richards solver. However, both
models clearly overestimated top soil wetting compared to observations. An asset of the
particle based approach is that the assumption of local equilibrium equation during infiltration
may be easily ignored. Specifically we did this to less the idea whether bypassing of a fast
water fraction might explain the model bias in the topsoil. To this end infiltrating event water
particles were treated as second particle type, which travel initially mainly gravity driven in
the largest pore fraction at maximum drift, and yet experience a slow diffusive mixing with
the pre-event water particles within a characteristic mixing time. Simulations with the particle
model in the non-equilibrium mode performed evidently distinctly different in the topsoil, and
were rather sensitive to the diffusion coefficient $D_{mix}$ describing mixing of event water
particles. When assuming $D_{mix}$ equal to the 40% quantile of the D-(θ) curve, the mixed water
fraction of the particle model was in good accordance with observed soil moisture changes
at0.025 and 0.1 m depths after the rainfall and at the end of the simulation period.

Our findings are in line with the early findings of Ewen (1996b). The diffusive mixing term
parameter $D_{mix}$ is perhaps easier to interpret as the λ parameter Ewen (1996b) introduced to
account for displacement of old water by new water particles, notwithstanding that
displacement of pre-event water seems to play a key role in feeding macropore flow (Klaus et
al, 2013; Klaus et al., 2014). Contrary to the exponential mixing term Davis and Beven (2012)
introduced to stop rapid flow in the MIP model, we used a uniform distribution which
maximizes entropy of the mixed particles (Klaus et al., 2015).

### 5.3 Conclusions and Outlook

We conclude overall that the proposed non-linear random walk of water particles is an
interesting stochastic alternative for simulating soil moisture dynamics in the unsaturated
zone, which preserves the influence of capillarity and makes use of established soil physics.
The approach is easy to implement, even in two or three dimensions and fully mass
conservative. The drawback is the required high density of particles, arising from the small


ratio of event water to pre-event water in soil, which might become a challenge when working in larger domains and several dimensions. However, due to its simplicity the model is straight forward to implement on a parallel computer.

The particle approach is particular interesting, as the implementation of fast non-equilibrium infiltration and the separation of event and pre-event water is straight forward, compared to for instance a non-local formulation of the Richards equation (Neuweiler et al., 2012). In line with Ewen (1996) we hence regard particle based models as particularly promising to deal with preferential transport of solutes (optionally also heat), and to explore transit time distributions in a forward mode.

We are aware, that the evidence we provided here is a somewhat tentative first step corroborate the flexibility of the particle based approach to include non-equilibrium flow and matrix flow in the same stochastic, physical framework. A much more exhaustive treatment of this issue is provided in a forthcoming study which presents and extension of the concept to a 2 dimensional domain with topologically explicit macropores and the test of concurring hypothesis to represent infiltration into macropores as well as macropore matrix interactions.

ACKNOWLEDGMENTS

This study contributes to and greatly benefited from the "Catchments As Organized Systems" (CAOS) research unit. We sincerely thank the German Research Foundation (DFG) for funding (FOR 1598, ZE 533/9-1). The authors acknowledge support by Deutsche Forschungsgemeinschaft and the Open Access Publishing Fund of Karlsruhe Institute of Technology (KIT). The service charges for this open access publication have been covered by a Research Centre of the Helmholtz Association. The code and the simulation projects underlying this study are freely available on request.





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




**7   FIGURES**

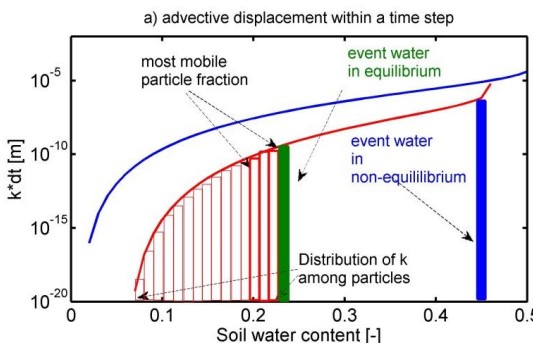

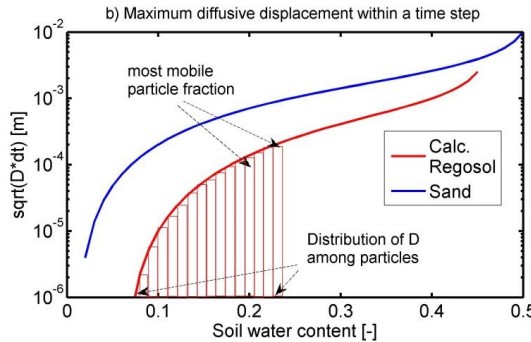


Figure 1: Advective/drift displacement of a particle k($\theta$) dt (panel a) and maximum diffusive
displacement (D($\theta$)dt)$^{0.5}$ (panel b) plotted against soil water content for the sand on limestone
in the Attert catchment and the Calcaric Regosol on loess in the Weiherbach catchment. The
vertical bars visualize the distribution of the D among the particles, representing water in
different pore size fractions. The arrows mark the most mobile particle fraction in the five
upper soil moisture classes. The red and the blue rectangle highlight the case when treating
event water either as in local equilibrium and particles travel according to D(($\theta$(t+0.5$\Delta$t)) and
k(($\theta$(t+0.5$\Delta$t)) or when they enter the coarsest pores and travel according $k_s$.



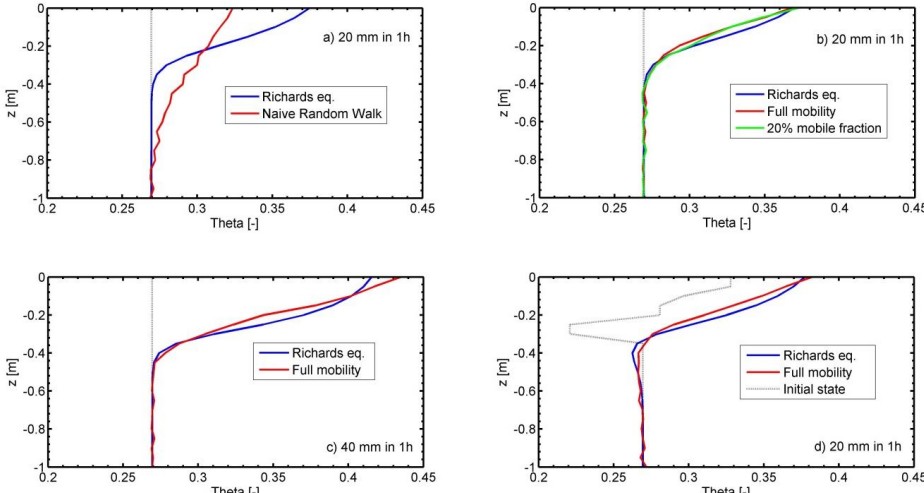


Figure 2: Final soil moisture profiles simulated for the sandy soil with the naive random walk
(panel a) and different versions of the particle model compared to the Richards equation for
two different block rains starting from the uniform initial state (panel c and b), and 20 mm
block rain on the s-shaped initial state (panel c). The dashed grey line marks the initial soil
moisture profiles.






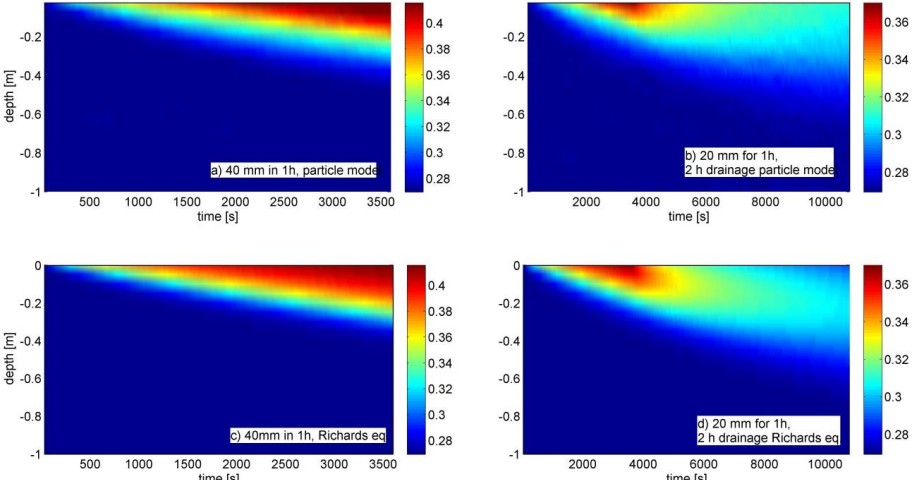


Figure 3: Time series of soil moisture simulated with the particle model and the Richards
solver for the sandy soil as 2d color plots for a simulated wetting event of 40 mm in 1 h (panel
a) particle model and c) Richards' solver) and a 1 h wetting and a 2 h drainage period (panel
b) particle model and d) Richards' solver).





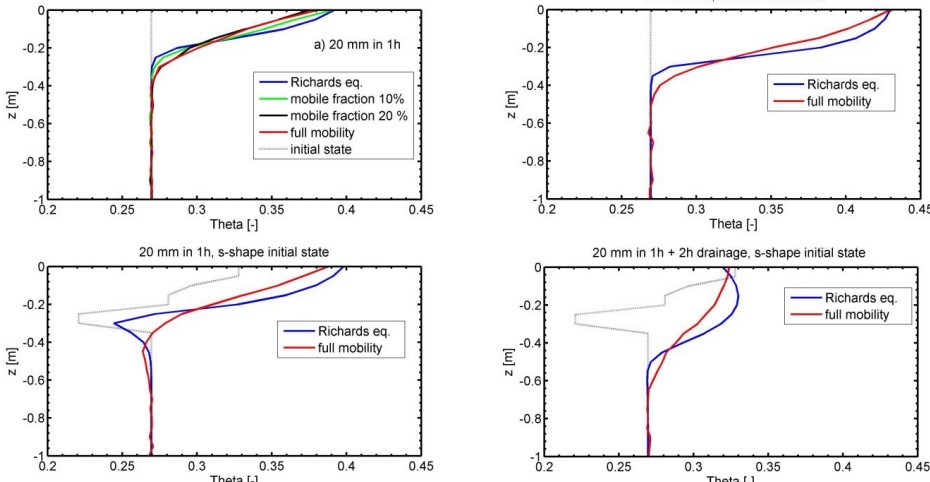


Figure 4: Final soil moisture profiles simulated for the young silty soil on schist. Panel a)
compares simulations with the full mobility particle model as well as versions assuming
mobile fractions of 10% or 20 %, respectively 20%. The remaining panels compare the full
class approach against the Richards equation starting from a uniform initial state after a 40
mm block rain (panel b), and from an s-shaped initial state after 20 mm of rainfall in 1 h
(panel c) and a subsequent drainage phase of 2h (panel d). The dashed grey line marks the
initial soil moisture profiles.



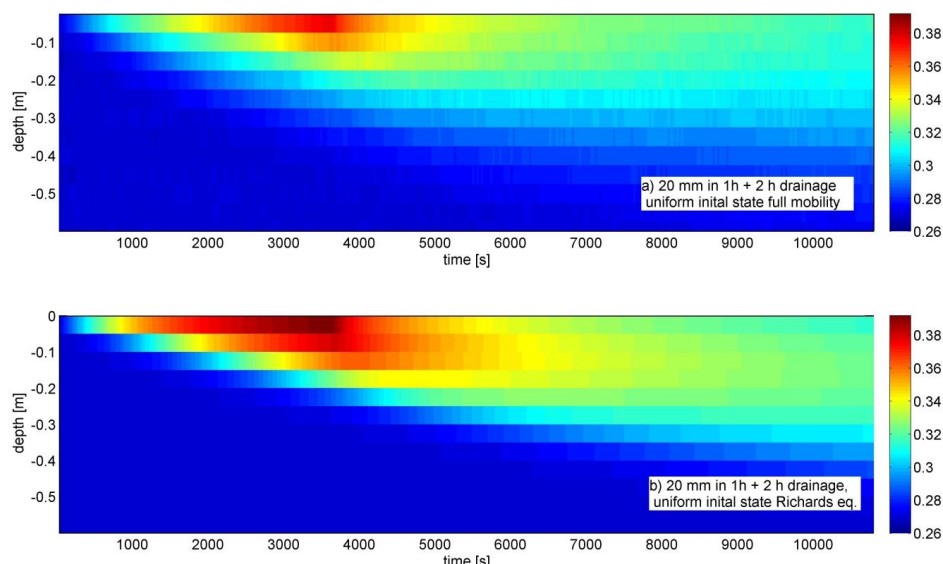


Figure 5: Time series simulated soil moisture profiles in the upper 60 cm: the full mobility
particle model in panel a) and the Richards solver in panel b) for the young silty soil on schist
for 20 mm infiltration of the uniform initial state and a subsequent drying of 2 hours.





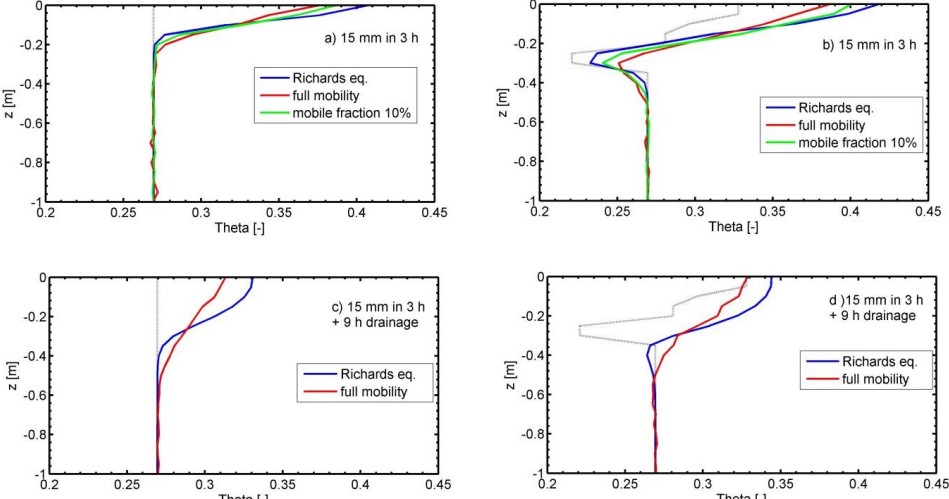


Figure 6: Final soil moisture profiles simulated for Calcaric Regosol on loess. Panels a) and b)
compare the particle model in the full mobility model (solid red) and in a mobile fraction of
10 % (solid green) to the Richards solver for 15 mm rainfall input in 3h. Panels c) and d)
compare the Richards solver and the particle model after 15 mm infiltration in 3 h and a
subsequent drainage phase of 9 h. The dashed grey line marks the initial soil moisture
profiles.





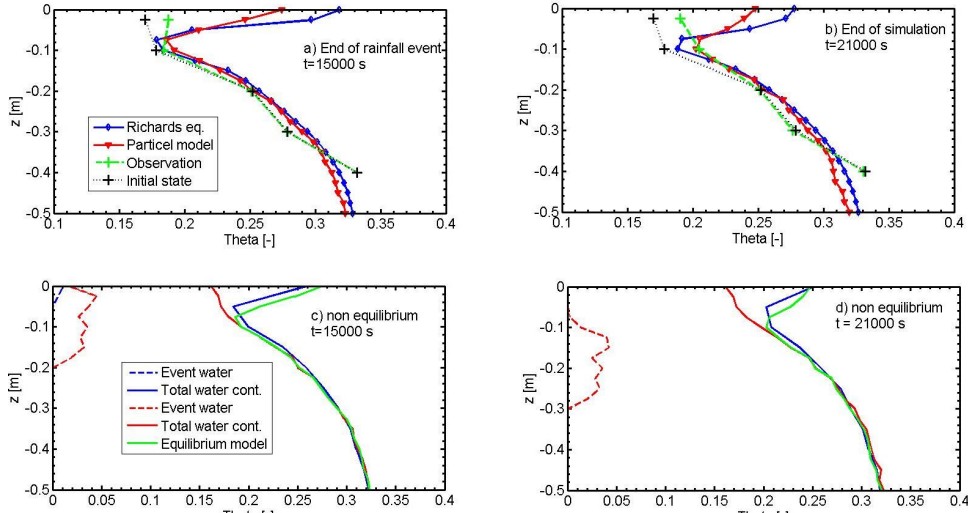


Figure 7: Soil moisture profiles simulated with the Richards equation (solid blue) and the particle model (in the full mobility mode assuming equilibrium infiltration) compared to observations in different depths at the end of the precipitation event (panel a), 15000s) and the end of simulation (panel b), 21000s). Initial soil moisture observations are given as black, intermediate and final observations as green crosses. Panels c) and d) present fractions of event water (dashed lines) total water content (pre-event + mixed water) for simulations assuming non-equilibrium infiltration. Blue lines correspond to $t_{mix} = 4300s$, red lines to $t_{mix} = 7300s$, the solid green line shows the soil water content simulated with equilibrium infiltration.

705





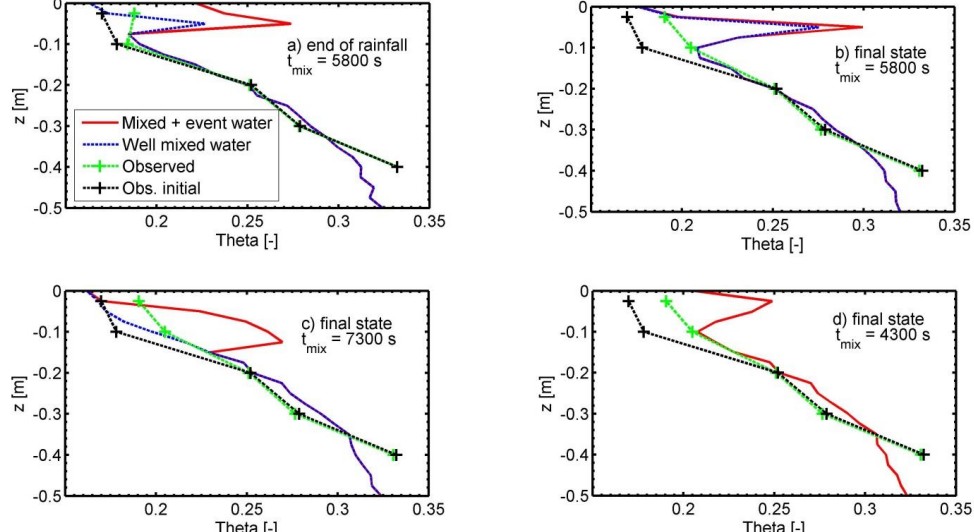

706

Figure 8: Non equilibrium simulations compared against observed soil moisture values, for

$t_{mix}$ = 5800s after the rainfall event (panel a) and at the end of simulation (panel b). Panel c)

and d) present the final state for $t_{mix}$ = 7300s or $t_{mix}$ = 4300s, respectively.

710



## 8 TABLES

Table 1: Soil hydraulic parameters of the sandy soil on limestone, the young silty soil on schist and the Calcaric Regosol on loess: saturated hydraulic conductivity $k_s$, saturated and residual water contents $\theta_s$, $\theta_r$, air entry value $\alpha$, shape parameter n.

| Soil type | $k_s$ [m/s] | $\theta_s$[-] | $\theta_r$[-] | $\alpha$[m$^{-1}$] | n[-] |
|---|---|---|---|---|---|
| **Sand on limestone** | $2.23*10^{-4}$ | 0.508 | 0.01 | 4.71 | 1.475 |
| **Young silty soil on schist** | $2.62\ 10^{-4}$ | 0.51 | 0.12 | 6.45 | 1.50 |
| **Calc. Regosol on loess** | $6.0\ 10^{-6}$ | 0.46 | 0.06 | 1.50 | 1.36 |






Table 2: Characteristics of the numerical benchmarks: rainfall input P, initial condition $\theta_{ini}$,
simulation time $t_{sim}$

| Soil type | Wetting | Wetting | Wetting | Wetting &drying |
|---|---|---|---|---|
| **Sand** | P =20 mm in 1h<br>$\theta_{ini}$ = uniform<br>$t_{sim}$ =1h | P =40 mm in 1h<br>$\theta_{ini}$ = uniform<br>$t_{sim}$ =1h | P =20 mm in 1h<br>$\theta_{ini}$ = s-shape<br>$t_{sim}$ =1h | P =20 mm in 1h<br>$\theta_{ini}$ = uniform<br>$t_{sim}$ =3h |
| **Silty soil** | P =20 mm in 1h<br>$\theta_{ini}$ = uniform<br>$t_{sim}$ =1h | P =40 mm in 1h<br>$\theta_{ini}$ = uniform<br>$t_{sim}$ =1h | P =20 mm in 1h<br>$\theta_{ini}$ = s-shape<br>$t_{sim}$ =1h | Input: 20 mm in 1h<br>initial con.: uniform<br>Duration: 2h |
| **Calc. Regosol** | P =20 mm in 1h<br>$\theta_{ini}$ = s-shape<br>$t_{sim}$ =1h | P =20 mm in 4h<br>$\theta_{ini}$ = uniform<br>$t_{sim}$ =4h | P =15 mm in 3h<br>$\theta_{ini}$ = s-shape<br>$t_{sim}$ =3h | P =15 mm in 3h<br>$\theta_{ini}$ = uniform<br>$t_{sim}$ =6h |







Table 3: Top soil and the subsoil hydraulic properties at the central meteorological station in
the Weiherbach catchment: saturated hydraulic conductivity $k_s$, saturated and residual water
contents $\theta_s$, $\theta_r$, air entry value $\alpha$, shape parameter n.

| Depth [ m] | $k_s$ [m/s] | $\theta_s$[-] | $\theta_r$[-] | $\alpha$[m$^{-1}$] | n[-] |
|---|---|---|---|---|---|
| 0 - 0.3 | $6.0 \times 10^{-6}$ | 0.46 | 0.06 | 1.50 | 1.36 |
| > 0.3 | $3.4 \times 10^{-6}$ | 0.44 | 0.06 | 1.50 | 1.36 |



