# Peer review of "A Lagrangian model for soil water dynamics during rainfall driven conditions"

_Hydrology and Earth System Sciences, 2016_

## Referee Comment (RC1) · Anonymous Referee #1 · 17 May 2016

The paper proposes a stochastic random walk particle tracking based approach for the modeling of soil-water dynamics in the unsaturated zone. First, I find this an interesting approach that is conceptually relatively simply despite the non-linearity (saturation-dependence) of the drift and diffusion coefficients, which impacts on the numerical efficiency of the scheme. The proposed approach is validated against numerical (finite difference?) solutions of the Richards equation and to a real-world benchmark. The random walk model does not fit perfectly with the finite-difference(?) solution of the Richards equation, the possible conceptual reasons of which will be discussed in the following.

The stochastic approach describes the motion of fluid particles by a stochastic differen-

tial equation, whose equivalent Fokker-Planck equation resembles (but is not equal to) the Richards equation. Equation (2) of the paper is not equivalent to Eq. (1). Rather, it corresponds to the Fokker-Planck equation (or convection-diffusion equation due to the diffusion correction), see, for example the textbook by Risken (The Fokker-Planck Equation),

(*) del \theta/del t = del/del z [k(theta) theta] + del/del z [D(theta) del/del z theta].

The difference between this Fokker-Planck equation and the Richards equation (1) in the manuscript is the advection term. In equation (1) it is k(theta) while in the above Fokker-Planck equation it is - del/del z [k(theta) theta] (Note that by convention the advection term in the Fokker-Planck equation is as in del theta /del t = - del / del z u theta + ...).

If we use the following more standard form (e.g., Eq. [9.4.100] in Bear, Dynamics of Fluids in Porous Media) of the Richards equation

del theta /del t = del/del z [K(theta) + D(theta) del/del z theta],

we can identify an advection term if we write it as

(**) del theta /del t = K'(theta) del/del z theta + del/del z [D(theta) del/del z theta]

where K' is the derivative of K with respect to theta.

The Richards equation in this form, however, is not a Fokker-Planck equation because it is not in a divergence-form (compare the advection terms in (*) and (**)). In order to achieve this one could

(i) consider the new variable

theta' = del theta/del z, which satisfies the Fokker-Planck equation

del theta' /del z = del /del z [K'(theta) theta'] + del^2/del z^2 D(theta) theta'.

The corresponding Langevin equation is given by

d z = - K'(theta) dt + Z sqrt{6 D(theta) dt},

where z now is adjoint to the saturation gradient.

or

(ii) define the velocity u(theta) = K(theta)/theta (see also the paper by Zoia et al., PRE 81,031104, 2010)

This may be the more suitable definition for the purpose of the manuscript. Using this formulation, the Richards equation becomes the Fokker-Planck equation

del theta/del t = del/del z [u(theta) theta + D(theta) del/del z theta]

or equivalently

del theta/del t = del/del z [u(theta) theta - D'(theta) theta + del/del z D(theta) theta]

with D'(theta) = d D(theta)/d theta. This Fokker-Planck equation is equivalent to the Langevin equation

dz = - u(theta)dt - D'(theta)dt + Z sqrt{6 D(theta)dt}

In summary, my main concern arises from Eq. (1) and (2), which are not equivalent. If the implementation of the random walk method is indeed based on Eq. (2) with the drift terms specified as -k(theta), the random walk method is not equivalent to solving the Richards equation, but another problem, whose physical meaning is not clear. Thus, I encourage the authors to revise the manuscript along the lines detailed above.
* * *

---

## Referee Comment (RC2) · Anonymous Referee #2 · 30 May 2016

The authors compare two random walk approaches to simulate Richard's equation. One is straightforward (called naïve) and the author is based on the fact that particles are distributed among the different capillary tubes given by the retention curve. The latter is a nice approach worth of publication. I am missing technical details on the applicability of these two methods in the paper in regards to the gradient of dispersion. I do not see how it is estimated. Also, an explanation of why the first method does not work. In theory, one should expect that the naïve method works for vary large number of particles? What are we missing here? Please, explain exactly how the two methods are implemented step by step.

---

## Referee Comment (RC3) · Anonymous Referee #3 · 1 Jun 2016

The paper deals with water flow in unsaturated soils simulated by a particle based model. The authors used the water content based formulation of the Richards equation to define an equivalent Fokker Planck equation. If the link between the PDE (eq. 1) and the stochastic equation (SE- eq.2) has been demonstrated for linear problems like the advection-dispersion equation, the analogy for highly nonlinear problem is fully intuitive, as well as the nice and smart implementations of the particle based method. Moreover, the easy way to extend the SE to preferential flow and mixing makes the particle based method very attractive as shown by the simulation of field data. As stated by the authors, this work is a first step in the development of this approach.

The paper needs some revisions according to the following comments:

[Figure]

1. It should be clearly stated that the chosen Richards formulation cannot be applied to heterogeneous domains.

2. The time varying parameters are handled using a predictor-corrector scheme which consists in computing the parameters at time t+0.5 ïĄĎt. The parameters values at t+0.5 ïĄĎt are not representative of the equivalent parameters defined over ïĄĎt due to the strong non linearity of the parameters with respect to the variables. It may be a good approximation for small time steps. Since the optimal time step is not known a priori, this makes the scheme tricky from a numerical point of view.

3. The comparison with the three theoretical benchmarks is quite convincing for short times. However, the difference between the PDE formulation and SE increases with time (fig. 3, b&d, fig 5). Is it due to drainage? Is there a bias in the method? An additional simulation with drainage of an initial saturated soil may provide some information. This point is critical. It is probably not possible to demonstrate mathematically that PDE and SE are equivalent. Therefore, detailed numerical experiments are required. At least, the authors should also provide a long term simulation (over one year) with time varying boundary conditions (infiltration, evaporation) and compare the SE to a reference solution (fine time and space discretizations) obtained by Richards equation.

4. Both approaches are also compared on a set of field data. Since the full mixing particle model differs from the Richards model by its formulation only, one model cannot be better than the other as it is stated lines 359 to 366 (p. 12). The difference is only due to the mathematical and numerical approximations made to establish the SE and to solve it.

---

## Author Comment (AC1) · 3 Aug 2016

**Reviewer 1:** The paper proposes a stochastic random walk particle tracking based approach for the modeling of soil-water dynamics in the unsaturated zone. First, I find this an interesting approach that is conceptually relatively simply despite the non-linearity (saturation dependence) of the drift and diffusion coefficients, which impacts on the numerical efficiency of the scheme. The proposed approach is validated against numerical (finite difference?) solutions of the Richards equation and to a real-world benchmark.

**Erwin Zehe (EZ)**: We sincerely thank the anonymous reviewer for his encouraging comments and helpful comments. Yes we use centered finite differences and will mention this in the revised manuscript.

**Reviewer 1:** The random walk model does not fit perfectly with the finite-difference solution of the Richards equation, the possible conceptual reasons of which will be discussed in the following. The stochastic approach describes the motion of fluid particles by a stochastic differential equation, whose equivalent Fokker-Planck equation resembles (but is not equal to) the Richards equation. Equation (2) of the paper is not equivalent to Eq. (1). Rather, it corresponds to the Fokker-Planck equation (or convection-diffusion equation due to the diffusion correction), see, for example the textbook by Risken (The Fokker-Planck Equation), (*) del ntheta/del t = del/del z [k(theta) theta] + del/del z [D(theta) del/del z theta. The difference between this Fokker-Planck equation and the Richards equation (1) in the manuscript is the advection term. In equation (1) it is k(theta) while in the above Fokker-Planck equation it is - del/del z [k(theta) theta] (Note that by convention the advection term in the Fokker-Planck equation is as in del theta /del t = - del / del z u theta + ...).
If we use the following more standard form (e.g., Eq. [9.4.100] in Bear, Dynamics of Fluids in Porous Media) of the Richards equation del theta /del t = del/del z [K(theta) + D(theta) del/del z theta], we can identify an advection term if we write it as (**) del theta /del t = K'(theta) del/del z theta + del/del z [D(theta) del/del z theta] where K' is the derivative of K with respect to theta.

**(EZ)**: We sincerely thank for this important point. Eq. 1 in the first manuscript version is in fact wrong, as it is not the correct form of the Richards equation:

$$\frac{\partial \theta}{\partial t} = -k(\theta) + \frac{\partial}{\partial z}\left( D(\theta)\frac{\partial \theta}{\partial z}\right)$$

$$D(\theta) = k(\theta)\frac{\partial \psi}{\partial \theta}$$

The correct version of the Richards equation in the soil moisture base form is, as correctly pointed out by the Reviewer:

$$\frac{\partial \theta}{\partial t} = \frac{\partial k(\theta)}{\partial z} + \frac{\partial}{\partial z}\left( D(\theta)\frac{\partial \theta}{\partial z}\right)$$

$$D(\theta) = k(\theta)\frac{\partial \psi}{\partial \theta}$$

We sincerely apologize for that and will revise the manuscript accordingly.

**Reviewer 1**: The Richards equation in this form, however, is not a Fokker-Planck equation because it is not in a divergence-form (compare the advection terms in (*) and (**)). In order to achieve this
- one could (i) consider the new variable theta' = del theta/del z, which satisfies the Fokker-Planck equation
  del theta' /del z = del /del z [K'(theta) theta'] + del^2/del z^2 D(theta) theta'. The

corresponding Langevin equation is given by d z = - K'(theta) dt + Z sqrt{6 D(theta) dt},where z now is adjoint to the saturation gradient.

- Or (ii) define the velocity u(theta) = K(theta)/theta (see also the paper by Zoia et al., 2010). This may be the more suitable definition for the purpose of the manuscript. Using this formulation, the Richards equation becomes the Fokker-Planck equation. del theta/del t = del/del z [u(theta) theta + D(theta) del/del z theta] or equivalently del theta/del t = del/del z [u(theta) theta - D'(theta) theta + del/del z D(theta) theta] with D'(theta) = d D(theta)/d theta. This Fokker-Planck equation is equivalent to the Langevin equation dz = - u(theta)dt - D'(theta)dt + Z sqrt{6 D(theta)dt}

**EZ:** Again we thank for this important point. We prefer the second formulation (ii) and will revise the manuscript accordingly.

$$\frac{\partial \theta}{\partial t} = \frac{\partial}{\partial z}\left[\frac{k(\theta)}{\theta}\theta\right] + \frac{\partial}{\partial z}\left(D(\theta)\frac{\partial \theta}{\partial z}\right)$$

Or equivalently

$$\frac{\partial \theta}{\partial t} = \frac{\partial}{\partial z}\left[\frac{k(\theta)}{\theta}\theta - \frac{\partial D(\theta)}{\partial z}\theta\right] + \frac{\partial^2}{\partial z^2}(D(\theta)\theta).$$

**Reviewer 1:** In summary, my main concern arises from Eq. (1) and (2), which are not equivalent. If the implementation of the random walk method is indeed based on Eq. (2) with the drift terms specified as -k(theta), the random walk method is not equivalent to solving the Richards equation, but another problem, whose physical meaning is not clear. Thus, I encourage the authors to revise the manuscript along the lines detailed above.

**EZ**: We already implemented the corrected version of the Langevin Equation.

$$z(t+\Delta t) = \left(-\frac{k(\theta(t))}{\theta(t)} - \frac{\partial D(\theta(t))}{\partial z}\right)\cdot \Delta t + Z\sqrt{6\cdot D(\theta(t))\cdot \Delta t}$$

It yields simulations in much better accordance with the Richards solver as the former version, particularly when it is operated using normally distributed random numbers in case of uniformly distributed ones.

$$z(t+\Delta t) = \left(-\frac{k(\theta(t))}{\theta(t)} - \frac{\partial D(\theta(t))}{\partial z}\right)\cdot \Delta t + \xi\sqrt{2\cdot D(\theta(t))\cdot \Delta t}$$

$$\xi \in N(0,1)$$

Figure 1 corroborates for the sandy soil that the improved particle model matches the Richards solver much better than the old version, particularly for a reduced grid size of 2.5 cm the matching is almost perfect.

[Figure]

Figure 1: Comparison of the old particle model and the new one based on the corrected form of the Langevin Equation using N(0,1) random numbers.

With respect to the physical meaning the proposed particle model remains of course an effective approach, as water does in fact flow as a continuum through the pore space. The main asset is however, that it still makes use of established soil physics, and that it allows for a straight forward implementation of non-equilibrium preferential flow. In fact the core objective of the study is to propose an alternative to the Richards equation for the case of rainfall driven conditions, which may cope with preferential flow and in an alternative manner and which avoids challenges of continuum approaches as for instance the dealing with partly wetted macropores or with macropore flow that hits the closed end of a macropore. We think the ultimate model is a hybrid, which uses the particle approach during rainfall driven conditions, when time stepping needs to be in the order of minutes, due to the characteristic time scale of changes in rainfall intensity. The primary variable is thus soil moisture/particle density. During radiation driven conditions when water flow is slow and in local equilibrium, it is favorable to switch to a Richards solver, because it works well and it is much more computationally efficient and treatment of for instance root water uptake is much more straightforward. We will stress this point in the revised manuscript.

We again thank Reviewer 1 for her/his insightful comments that clearly helped us to improve the propose model.

Erwin Zehe

---

## Author Comment (AC2) · 3 Aug 2016

**Reviewer 2:** The authors compare two random walk approaches to simulate Richard's equation. One is straightforward (called naïve) and the author is based on the fact that particles are distributed among the different capillary tubes given by the retention curve. The latter is a nice approach worth of publication.

**Erwin Zehe (EZ):** We sincerely thank the anonymous reviewer for his encouraging comments and helpful comments.

**Reviewer 2:** I am missing technical details on the applicability of these two methods in the paper in regards to the gradient of dispersion. I do not see how it is estimated. Also, an explanation of why the first method does not work. In theory, one should expect that the naïve method works for vary large number of particles? What are we missing here? Please, explain exactly how the two methods are implemented step by step.

**EZ:** The spatial gradient of the dispersion coefficient is estimated by means of a centered finite difference. This is straight forward, as the dispersion coefficient is in both approaches well defined in each grid box. We will better explain this point in the revised manuscript.

Based on the widespread evidence that particle tracking is suitable for simulating solute transport, we also expected the "naïve" approach to work well for large particle numbers and a suitable updating rate of the dispersion coefficient. We started initial simulations with the naïve approach with $10^4$ particles and found this approach systematically too overestimates depletion of gradients and the thus vertical redistribution of water compared to the Richards model. Neither an increase of the particle number to $10^6$ nor an iterative updating of the dispersion coefficient nor shorter time steps fixed the problem.

Within the naïve approach all water particles in the pore space experience the same dispersion $D(\theta(t))$. This assumption is based on the analogy to the solute transport problem, where all solute particles in a flow field experience indeed the same dispersion: they experience so to say the same "average path length", which implies that the diffusive step scales for all solutes with the same dispersion coefficient $D^{solute}$ as follows $Z\sqrt{6 \cdot D^{solute} \cdot \Delta t}$. This is however not the case for water parcels/molecules in porous media, because diffusive flow velocities decrease with decreasing pore size. To account for this the diffusive step **cannot** scale for all particles with same maximum $D(\theta(t))$, it needs to reflect the distribution of D within the different wetted pore sizes fraction. To achieve this we subdivide the particles in a grid cell into N bins (800) and calculate D starting from the residual moisture content to the $\theta_r$ stepwise to $\theta(t)$ using a step with $\Delta\theta = (\theta(t) - \theta_r)/N$. The diffusive step for particles within bin i scales $Z\sqrt{6 \cdot D(\theta_r + i \cdot \Delta\theta) \cdot \Delta t}$, for i=1 … N. We will add a similar explanation to the revised manuscript.

Fortunately Reviewer 1 pointed out that our implementation of the Langevin Equation was not entirely correct (see our corresponding reply). Figure 1 shows a simulation with the corrected model in the full class mode (using 200 bins) and the "naive" approach, also based on the correct form of the Langevin equation. The full class model shows a nearly perfect match of the Richards solver, while the naïve approach shows the above explained deficit.

[Figure]

 Figure 1: Comparison of the new particle model based on the corrected form of the Langevin Equation using N(0,1) random numbers in the full class mode, with the naïve model based on the corrected Langevin Equation.

We again thank Reviewer 2 for her/his insightful comments that will surely help us to improve the presentation quality of our study.

Erwin Zehe

---

## Author Comment (AC3) · 3 Aug 2016

**Reviewer 3:** The paper deals with water flow in unsaturated soils simulated by a particle based model. The authors used the water content based formulation of the Richards equation to define an equivalent Fokker Planck equation. If the link between the PDE (eq. 1) and the stochastic equation (SE- eq.2) has been demonstrated for linear problems like the advection-dispersion equation, the analogy for highly nonlinear problem is fully intuitive, as well as the nice and smart implementations of the particle based method. Moreover, the easy way to extend the SE to preferential flow and mixing makes the particle based method very attractive as shown by the simulation of field data. As stated by the authors, this work is a first step in the development of this approach.

**Erwin Zehe (EZ)**: We sincerely thank the anonymous reviewer for his encouraging comments and helpful comments. In fact the core objective of the study is to propose an alternative to the Richards equation for the case of rainfall driven conditions, which may cope with preferential flow and in an alternative manner and which avoids challenges of continuum approaches as for instance the dealing with partly wetted macropores or with macropore flow that hits the closed end of a macropore. The particle approach allows a straightforward accounting of a fast flow component by treating the event water as second particle fraction. We agree that we just presented a first tentative feasibility study in this respect, which is promising.

The scope of the particle model is thus clearly on infiltration and soil water dynamics during rainfall driven conditions. We think the ultimate model is a hybrid, which uses the particle approach during rainfall driven conditions, when time stepping needs to be in the order of minutes, due to the characteristic time scale of changes in rainfall intensity. The primary variable is thus soil moisture/particle density. During radiation driven conditions when water flow is slow and in local equilibrium, it is favorable to switch to a Richards solver, because it works well and it is much more computationally efficient and treatment of for instance root water uptake is much more straightforward. We will stress this point in the revised manuscript.

**Reviewer 3:** The paper needs some revisions according to the following comments:

**EZ**: Please note that we address the reviewers point 3 first.

**Reviewer 3:** The comparison with the three theoretical benchmarks is quite convincing for short times. However, the difference between the PDE formulation and SE increases with time (fig. 3, b&d, fig 5). Is it due to drainage? Is there a bias in the method? An additional simulation with drainage of an initial saturated soil may provide some information. This point is critical. It is probably not possible to demonstrate mathematically that PDE and SE are equivalent.

**EZ**. Our model was indeed biased. Equation 1) in the original manuscript is not the Richards equation! When starting with the correct version

$$\frac{\partial \theta}{\partial t} = \frac{\partial k(\theta)}{\partial z} + \frac{\partial}{\partial z}\left( D(\theta) \frac{\partial \theta}{\partial z} \right)$$

$$D(\theta) = k(\theta) \frac{\partial \psi}{\partial \theta}$$

This may be rewritten as

$$\frac{\partial \theta}{\partial t} = \frac{\partial}{\partial z}\left[ \frac{k(\theta)}{\theta} \theta \right] + \frac{\partial}{\partial z}\left( D(\theta) \frac{\partial \theta}{\partial z} \right)$$

Or equivalently

$$\frac{\partial \theta}{\partial t} = \frac{\partial}{\partial z} \left[ \frac{k(\theta)}{\theta} \theta - \frac{\partial D(\theta)}{\partial z} \theta \right] + \frac{\partial^2}{\partial z^2} \left( D(\theta)\theta \right).$$

The latter is equivalent to the Fokker Planck Equation. The corresponding Langevin Equation is

$$z(t + \Delta t) = \left( -\frac{k(\theta(t))}{\theta(t)} - \frac{\partial D(\theta(t))}{\partial z} \right) \cdot \Delta t + Z\sqrt{6 \cdot D(\theta(t)) \cdot \Delta t}$$

, which differs from the equation (2) in the current manuscript. We already implemented the corrected version of the Langevin Equation. It yields simulations in much better accordance with the Richards solver as the former version, particulary when it is operated using normally distributed random numbers in case of uniformly distributed ones.

$$z(t + \Delta t) = \left( -\frac{k(\theta(t))}{\theta(t)} - \frac{\partial D(\theta(t))}{\partial z} \right) \cdot \Delta t + \xi\sqrt{2 \cdot D(\theta(t)) \cdot \Delta t}$$

$$\xi \in N(0,1)$$

Figure 1 corroborates for the sandy soil that the improved particle model matches the Richards solver much better than the old version, particularly for a reduced grid size of 2.5 cm the matching is almost perfect.

[Figure]

Figure 1: Comparison of the old particle model and the new one based on the corrected form of the Langevin Equation using N(0,1) random numbers (simulation time step was 20 s).

**Reviewer 3:** Therefore, detailed numerical experiments are required. At least, the authors should also provide a long term simulation (over one year) with time varying boundary conditions (infiltration, evaporation) and compare the SE to a reference solution (fine time and space discretizations) obtained by Richards equation.

**EZ:** We agree that additional simulations with time varying boundary conditions will further illustrate the validity of the approach and will provide additional long term simulations with transient rainfall conditions using observed rainfall events at the two sites, which cover the entire range of short term convective events up to stratiform long term events of several days duration. Please note that he scope of the particle model is thus clearly on infiltration and soil water dynamics during rainfall driven conditions, because of its potential to cope with preferential flow. We do **not** recommend the use of this type of model during radiation driven/fair weather conditions, because is offers no

advantage here. We thus think that a one year simulation is inappropriate, because this is not the scope of the propose model.

We think and will better explain that the ultimate model is a hybrid, which uses the particle approach during rainfall driven conditions, when time stepping needs to be in the order of minutes, due to the characteristic time scale of changes in rainfall intensity. The primary variable is thus soil moisture/particle density. During radiation driven conditions when water flow is slow and in local equilibrium, it is favorable to switch to a Richards solver, because it works well and it is much more computationally efficient and treatment of for instance root water uptake is much more straightforward. We will stress this point in the revised manuscript.

**Reviewer 3**. It should be clearly stated that the chosen Richards formulation cannot be applied to heterogeneous domains.

**EZ:** We will do so in the revised manuscript. Please note that the particle model is not compared against a solution of the Richards equation in the theta based form. We use a the Richards equation in the mixed form, which is to our notion the still the mass conservative benchmark for simulating soil water dynamics in the unsaturated zone in the absence of preferential flow. We will add additional test for system with different soil horizons.

**Reviewer 3:** The time varying parameters are handled using a predictor-corrector scheme which consists in computing the parameters at time t+0.5 ïA¸Dˇt. The parameters values at t+0.5 ïA¸Dˇt are not representative of the equivalent parameters defined over ïA¸Dˇt due to the strong non linearity of the parameters with respect to the variables. It may be a good approximation for small time steps. Since the optimal time step is not known a priori, this makes the scheme tricky from a numerical point of view.

**EZ:** This is a very good point. Figure 2 show simulations with the corrected particle model for the sandy soil using different constant time steps (20 s, 50 s, 100 s, 200 s, 500 s). Deviations for time steps of up to 100s are of order 0.5 % VOL, one may observe clear oscillations for time steps larger or equal than 200 s.

[Figure]

Figure 2: Simulations with the improved model using the the corrected form of the Langevin Equation for different constant time steps.

This implies that the particle model in fast draining soils needs to be operated at time steps of maximum 50 s (while larger time steps up to 600s are feasible for slower draining cohesive soils.). While thus appears as a drawback at first site time steps of 50s seconds are appropriate for the proposed model scope. Rainfall intensity changes at the scale of 1 – 6 minutes in case of convective rainfall events, which implies that the temporal resolution of highly resolved rainfall data does restrict the possible time to values smaller than 6 min (when treating with 6 minutes rainfall).

My personal experience is that a numerical accurate simulation of infiltration does, even when using a mass conservative implicit picard solver proposed by Celia et al. (1990) time step in the order of 10 seconds to assure convergence of the iteration. We will add similar tests by running the model at different time steps and discuss the issue restriction of the selected predictor corrector scheme per se, and in the light of the model scope.

**Reviewer 3**: Both approaches are also compared on a set of field data. Since the full mixing particle model differs from the Richards model by its formulation only, one model cannot be better than the other as it is stated lines 359 to 366 (p. 12). The difference is only due to the mathematical and numerical approximations made to establish the SE and to solve it.

**EZ**: This is absolutely correct we will reformulate this passage.

We again thank Reviewer 3 for her/his very much for the insightful comments that will surely help us to improve the quality of our study.

Erwin Zehe